# Phytochemistry, Medicinal Properties, Bioactive Compounds, and Therapeutic Potential of the Genus *Eremophila* (Scrophulariaceae)

**DOI:** 10.3390/molecules27227734

**Published:** 2022-11-10

**Authors:** Ian Edwin Cock, Linn Baghtchedjian, Marie-Elisabeth Cordon, Eléonore Dumont

**Affiliations:** 1Centre for Planetary Health and Food Security, Griffith University, Brisbane 4127, Australia; 2Ecole De Biologie Industrielle, 95800 Cergy, France

**Keywords:** Scrophulariaceae, Australian plant, emu bush, poverty bush, antibacterial activity, serrulatane diterpenoid

## Abstract

The genus *Eremophila* (family Scrophulariaceae) consists of approximately 200 species that are widely distributed in the semi-arid and arid regions of Australia. Multiple *Eremophila* spp. are used as traditional medicines by the First Australians in the areas in which they grow. They are used for their antibacterial, antifungal, antiviral, antioxidant, anti-diabetic, anti-inflammatory, and cardiac properties. Many species of this genus are beneficial against several diseases and ailments. The antibacterial properties of the genus have been relatively well studied, with several important compounds identified and their mechanisms studied. In particular, *Eremophila* spp. are rich in terpenoids, and the antimicrobial bioactivities of many of these compounds have already been confirmed. The therapeutic properties of *Eremophila* spp. preparations and purified compounds have received substantially less attention, and much study is required to validate the traditional uses and to highlight species that warrant further investigation as drug leads. The aim of this study is to review and summarise the research into the medicinal properties, therapeutic mechanisms, and phytochemistry of *Eremophila* spp., with the aim of focussing future studies into the therapeutic potential of this important genus.

## 1. Introduction

Prior to the advances in modern medicine that have occurred during the last century, traditional remedies were relied upon to treat a myriad of diseases, and this medicinal knowledge has been recorded for many civilisations globally. In contrast, the First Australians did not record their knowledge of Australian plants and instead relied on the oral transfer of this knowledge between generations. The general lack of written records has resulted in considerable loss of knowledge of First Australian traditional healing methods, or incorrect or incomplete documentation when recorded by European immigrants [1]. Considerable ethnobotanical, pharmacognostic, and pharmacological work is required to record and safeguard this knowledge for future generations.

As a result of its geographic isolation, Australian flora and fauna evolved separately from species in other regions, with minimal spread globally or minimal introduction of plants from other regions through most of the estimated 50,000 years that Australia has been populated by humans. This has resulted in one of the highest floral diversities and degrees of endemism globally. Furthermore, the size and wide array of Australian environmental conditions has resulted in plant species that have adapted to survive in these conditions. As plants produce secondary metabolites to assist in their survival, the diversity of unique plant species growing in harsh environments provides a largely untapped reservoir of natural products for drug discovery.

The First Australians have developed a detailed understanding of the medicinal potential of Australian flora through 50,000 years of trial and observation. Numerous plants were used therapeutically by individual tribal groups, and the ancestral knowledge has been transmitted orally from generation to generation. Unfortunately, few written accounts of the plant species used medicinally as well as of their preparation and application exist, with most records written from the perspective of European settlers. This has led to many dangerous or potentially fatal misrepresentations about the usage of plants by the First Australians. For example, Lassak and McCarthy [2] provide an account of the use of a native cycad fruit as a food. An early account by European settlers lists the fruit as a food without describing its preparation. Notably, the fruit contains substantial levels of toxic compounds. The First Australians placed the fruit in a stream of running water for several days before they considered it safe to consume. Reporting the fruit as a food without including details of its preparation resulted in several early settlers becoming seriously ill. An understanding of not only the species used by the First Australians, but also of their preparation and application methods is important for safe usage and also to highlight plant species worthy of further evaluation as drug leads.

Cold water decoctions were commonly used by the First Australians to prepare medicines [2]. As metal or clay containers were not readily available to the First Australians, it was difficult for them to boil water to prepare hot water infusions. Instead, most traditional medicines used by the First Australians were produced by crushing the plant material and soaking it in wooden bowls. The liquid was strained off, and the medicine could be drunk or used as a body wash. The plant material was also often pounded into a paste and mixed with animal fat to produce an ointment that was applied topically to the affected area. Alternatively, smoking ceremonies and smoke inhalation were also particularly widespread amongst First Australians, yet despite this, limited studies are yet to examine the effects of pyrolytic activity on the phytochemistry or therapeutic properties of Australian medicinal plants. One exception was a study that reported substantial phytochemical changes following the burning of *Eremophila longifolia* (R.Br.) F.Muell. leaves, as they would have been used in many traditional healing practices [3]. Instead, most laboratory-based studies have prepared and tested either essential oils or solvent extracts, and substantial further work is required to test First Australian medicines in the form similar to how they were traditionally used to validate their activity and to highlight new drug leads.

The taxonomy of the family Scrophulariaceae is complex and previously included approximately 5000 species from 275 genera. However, following reclassification, the family now consists of approximately 1830 species across 62 genera. A summary of the current phylogenetic classification of *Eremophila* within the family Scrophulariaceae is depicted in Appendix A. The genus *Eremophila* (Tribe Myoporeae) is one of the largest genera of Scrophulariaceae, consisting of more than 260 species, all of which are endemic to Australia [4,5,6]. The genus is widespread in arid areas of mainland Australia (none occur naturally in Tasmania), particularly in Western Australia, South Australia, and the Northern Territory. Whilst individual species can range in size from prostrate shrubs (e.g., *Eremohilia serpens* Chinnock) to small trees (e.g., *Eremophila bignoniflora* (Benth.) F.Muell.), most species grow as compact, low-growing shrubs. Figure 1a shows the growth form of *Eremophila maculata* (Ker Gawl.) F.Muell., which is characteristic of many species of this genus. The size and shape of *Eremophila* spp. is also variable, although the leaves are generally relatively small (as shown for *E. maculata* in Figure 1b) and may be shiny or hairy. The flowers also share common features across the genus, generally having five petals per flower, which are joined at their base, forming a tube (Figure 1b). The flower color can vary widely between species, with red-, purple-, lilac mauve-, cream-, white-, or even green-coloured flowers being common across many species. Notably, many *Eremophila* spp. share common names, which include emu bush or poverty bush, which relate to the belief that emus eat the fruit of those species and to the arid/poor environment in which they grow. Alternatively, multiple *Eremophila* spp. are also known as fuchsia bush due to the superficial resemblance of their flowers to those of some *Fuchsia* spp.

The First Australians had substantial knowledge of *Eremophila* spp. and utilised multiple species medicinally. Indeed, an earlier study noted that of ~70 species that had been identified as being used therapeutically by First Australians from central Australia, approximately a third of those species belonged to either the *Eremophila* or *Acacia* genera [7]. Of these, *Eremophila alternifolia* R.Br. was highlighted because of its importance to the First Australians [6,8], and it is still considered to be one of the most important medicinal plants amongst central Australian Aboriginal populations.

As well as their medicinal uses, *Eremophila* spp. have a wide variety of other uses by the First Australians, including their use in cleansing ceremonies, in initiation rites, and in lining graves. Additionally, some *Eremophila* spp. produce substantial levels of resin that may have been used as a natural cement and sealant [9], although this remains to be conclusively verified. Several studies have previously reviewed the traditional, medicinal uses of *Eremophila* spp. and their phytochemistry, and these served as a good starting point for our study [4,5,6]. However, those reviews were published in 1993/1994 and are now outdated. Our study updates the earlier reviews to include the wealth of recent studies in this field. Or review focusses on the therapeutic properties of *Eremophila* spp. and on therapeutically important phytochemical constituents, with the aim of highlighting future studies into the medicinal potential of this important genus and on its phytochemical constituents.

## 2. Ethnopharmacology

The hostile environment and conditions of Australia led the First Australians to develop therapeutic methods using elements readily available to them in their environment. The local flora consists of a myriad of unique species, some of which have been exploited by the First Australians for their therapeutic properties. The genus *Eremophila* was widely used by the First Australian communities in all of the regions in which they grew to treat a wide variety of illnesses and medical complaints. Of the *Eremophila* spp., approximately twenty species were most frequently used as medicines [7]. Appendix A records and summarises the ethnobotanical knowledge and traditional therapeutic uses of *Eremophila* spp. as traditional medicines, whilst Appendix A summarises the studies undertaken to validate their therapeutic properties.

The use of *Eremophila* spp. preparations to treat bacterial infections is particularly well reported. *Eremophila alternifolia* R.Br., *Eremophila duttonii* F.Muell., *Eremophila freelingii* F.Muell., *Eremophila gilesii* F.Muell., *Eremophila latrobei* F.Muell., *Eremophila longifolia* (R.Br.) F.Muell., and *Eremophila sturtii* R.Br. were traditionally used as antiseptic/antibacterial therapies to treat wounds and skin infections. Decoctions prepared from the leaves of *E. alternifolia, E. duttoni, E. freelingi, E. gilesii, E. longifolia*, and *E. strutii* were used topically as a hot bath by the First Australians to treat skin infections [1,2]. Similarly, decoctions prepared using young *Eremophila bignoniiflora* leaves were applied topically to treat bacterial skin diseases and wound infections. Alternatively, the leaves and twigs were wrapped around the head for the treatment of sinusitis and to relieve nasal congestion. The leaves of *E. latrobei* were prepared as a decoction and used as an antibacterial mouthwash/gargle to relieve sore throats. Decoctions of the leaves of *E. longifolia* were also applied as an eye wash and antiseptic for ophthalmic problems [1,2].

The First Australians also prepared a body scrub by mixing *Eremophila dalyana* leaf decoctions with animal fat. This was applied directly to the chest to treat chest pain [2,7]. Similarly, decoctions prepared from *Eremophila duttonii*, *Eremophila elderi*, or *Eremophila latrobei* leaves were applied to the affected area to alleviate the symptoms of rheumatism [2]. Rheumatism was also treated by the topical application of *Eremophila gilesii* decoctions to the affected area. In contrast, *Eremophila maculata* leaves are applied directly to the body as a poultice to treat rheumatism and chest pain.

In addition to topical application, several *Eremophila* spp. preparations were consumed to treat bacterial diseases. This was particularly evident for the treatment of gastrointestinal bacterial infections. Decoctions of *Eremophila sturtii* and *Eremophila freelingii* leaves were consumed by several First Australian groups to fight gastrointestinal diseases and food poisoning [1,2]. Similarly, *Eremophila goodwinii* leaf decoctions were ingested to purge the digestive system and to improve gastrointestinal health. Interestingly, several *Eremophila* spp. were also noted for their laxative effects, which may in part be responsible for the use of *Eremophila* spp. decoctions to improve gastrointestinal health. *Eremophila bignoniiflora* decoctions were particularly noted for their laxative effects. Decoctions prepared from the fruit of this species were ingested to purge the gastrointestinal tract of microbial pathogens during severe gastrointestinal illness. *Eremophila cuneifolia* leaf decoctions we also consumed to treat headaches as well as to alleviate other aches and pains [1,2,6,10]. 

*Eremophila* spp. preparations were also used for the treatment of viral respiratory diseases. The use of *Eremophila* spp. decoctions by multiple First Australian groups to treat colds and influenza were particularly well reported [1,2,6,10]. *Eremophila* spp. decoctions were also useful in alleviating the symptoms of these diseases, including headaches. Whilst multiple species were used to treat viral respiratory diseases, the consumption *of Eremophila alternifolia, Eremophila fraseri, Eremophila freelingii*, and *Eremophila longifolia* leaf decoctions were most frequently reported for the treatment of colds. Additionally, *Eremophila sturtii* leaf infusions were used as hot baths and for steam inhalation for colds and influenza.

Several *Eremophila* species were also used by the First Australians as a body wash for the treatment of scabies infestations (caused by the ecto-parasite *Carcoptes scabiei* L.). In particular, *Eremophila dalyan, Eremophila latrobei*, and *Eremophila paisley* decoctions and infusions were reported to be effective against this parasite [2]. Whilst reports are scarce about the use of *Eremophila* spp. preparations against other ecto-parasites, it is possible that they may also be useful against lice, fleas, bedbugs, and mites, although this remains to be verified. 

## 3. Medicinal Properties and Therapeutic Effects

### 3.1. Antibacterial Activity

Multiple studies have screened *Eremophila* spp. extracts for medically relevant bioactivities. Of these, the antibacterial properties have been the most extensively reported. One study screened multiple species for antibacterial activity and reported an interesting trend [11]. That study noted a correlation between the relative levels of the antimicrobial compounds present in *Eremophila* spp. that produce a sticky, oily, or waxy resin layer on green leaves and branches. Therefore, the high-resin-producing *Eremophila* spp. should be prioritised as candidates for future studies evaluating the antimicrobial properties of *Eremohilia* spp. extracts and isolated compounds. That study reported that *Eremophila lucida* leaf extracts exhibited noteworthy antibacterial activity against several Gram-positive bacterial species, including several *Staphylococcus* and *Streptococcus* species. The authors subsequently used a bioactivity-driven separation approach to isolate three major constituents from the leaf resin. The resin was initially extracted with acetone, and the extract was then fractionated with hexane. The hexane fraction displayed good antibacterial activity against *Staphylococcus aureus* and was therefore used to isolate the sesquiterpenoid farnesal and the diterpenoid viscidane (5α-hydroxyviscida-3,14-dien-20-oic acid). Viscidane had particularly good antibacterial activity, with MIC values of 65 μg/mL recorded against several *S. aureus* strains.

Another study by the same group examined the antimicrobial activity of *E. alternifolia* leaf extracts and the isolated fractions using broth microdilution methods, and MIC and MBC values were reported [12]. The authors of that study identified the flavanones pinobanksin, pinobanksin-3-acetate, and pinobanksin-3-cinnamate as well as the serrulatan diterpene 8-hydroxyserrulat-14-en-19-oic acid as particularly promising. Of these components, all except pinobanksin had substantial antibacterial activity. Notably, the serrulatan diterpenoid component was identified as being widespread amongst *Eremophila* spp. and may therefore contribute to the antibacterial properties of those species. However, the most potent antibacterial activity was measured for pinobanksin-3-cinnamate against several *S. aureus* strains (MIC values 10–20 µM), including methicillin-resistant and biofilm-forming strains. In contrast, all of the isolated *E. alternifolia* leaf compounds were completely ineffective against the Gram-negative bacterium *Escherichia coli*. 

Similarly, a recent study reported that *Eremophila alternifolia* leaf extracts have substantial antifungal and antibacterial activity against multiple pathogens [13]. That study also demonstrated that the extracts inhibited several important cellular pathways and disrupted membrane integrity in Gram-positive bacterial species. The same study determined that several isolated *E. alternifolia* compounds have significant antifungal activity against the fungi *Cryptococcus gattii*, and *Cryptococcus neoformans* as well as against the yeasts *Candida albicans, Candida krusei*, and *Candida glabrata*.

Another study screened 39 plant species (including six *Eremophila* spp.) for antibacterial activity using an agar well-diffusion assay [14]. They reported that *Eremophila duttonii* had particularly good growth-inhibitory activity against *Bacillus cereus, Enterobacter faecalis, Staphylococcus aureus*, and *S. pyogenes*, with 10, 9, 12, and 14 mm inhibition zones measured, respectively. Notably, this was considered to be the best antibacterial activity measured for any of the plant extracts tested (on the basis of the measured inhibition zones). The authors also reported growth-inhibitory activity against *B. cereus* (albeit with substantially small inhibition zones). In contrast, all of the *Eremophila* spp. extracts (including *E. duttonii*) were completely ineffective against *Escherichia coli, Klebsiella pneumonia, Pseudomonas aeruginosa*, and *Salmonella typhimurium*, indicating that the extract components inhibit bacterial growth via selective mechanisms. That study did not quantify the potency of the extracts by determining MIC values for bacterial growth inhibition. Therefore, is it not possible to fully evaluate the antibacterial activity of the extracts nor to compare the activity with other studies. Notably, the choice of a solid phase diffusion assay may not have provided the best indication of the antibacterial activities of those extracts. The movement of molecules within solid-phase agar gels is dependent on the physiochemical characteristics of the molecules in the tested extracts. In general, large or low-polarity molecules diffuse poorly with agar gels [15], providing small inhibition zones that do not necessarily correspond to activity in biological systems. *Eremophila* spp. extracts are rich in terpenoids, which have relatively low solubility in aqueous solutions. It is likely that the bacterial-growth-inhibitory activity was substantially underestimated in that study due to the choice of assay, and further studies using liquid dilution-based MIC assays are required to confirm and quantify the antibacterial activity of these extracts. 

A different study used both disc diffusion assays and fluorescein diacetate (FDA) bactericidal assays to evaluate the growth-inhibitory activity of some diterpenoids isolated from *Eremophila sturtii* leaf extracts [16]. Unfortunately, that study focussed on the isolated compounds and did not test the crude extract for inhibitory activity. Thus, it is not possible to validate the traditional use of *E. sturtii* preparations against the tested bacterial species. However, noteworthy activity was reported for 3,8-dihydroxyserrulatic acid and serrulatic acid against *S. aureus*. Serrulic acid was a particularly good inhibitor of that bacterium, with a minimum bactericidal activity (MBC) of 15 μg/mL. A noteworthy although substantially higher MBC (200 μg/mL) was also determined for 3,8-dihydroxyserrulatic acid. However, both compounds were ineffective against *E. coli* and *P. aeruginosa* and against the yeast *Candida albicans*. As plant extracts often contain multiple compounds that may potentiate each other’s activity, testing only the isolated compounds without also testing the crude extract in parallel may have resulted in interesting therapeutic properties being overlooked, and future studies should also test *E. sturtii* solvent extractions against the same bacterial pathogens. 

A recent study reported that similar compounds isolated from *Eremophila glabra* leaf extracts also had noteworthy antibacterial activity [17]. The authors of that study isolated and screened seven serrulatan diterpenes and three flavonoids for antibacterial activity. All of the isolated compounds were tested against *Staphylococcus aureus* (NCTC 10442) and *Staphylococcus epidermidis* (ATCC 14990) using an agar diffusion antimicrobial test. Two flavonoids (hispidulin and jacesidin) as well as three diterpenoids (18-acetoxy-8-hydroxyserrulat-14-in-19-oic acid, 8,18,20-triacetoxyserrulat-14-in-19-oic acid, and 20-acetoxy-8-hydroxyserrulat-14-in-19-oic acid) displayed good growth-inhibitory activity against both bacteria, with minimum inhibitory concentrations (MICs) ranging from 32 to 512 μg/mL. However, whilst agar diffusion assays are suitable for polar flavonoid compounds, the lower solubility of the diterpenoids in aqueous solutions may have resulted in the potency of those compounds being underestimated. The MIC values of these compounds should be re-evaluated in more appropriate assays. Despite this, the serrulatan diterpene 20-acetoxy-8-hydroxyserrulat-14-en-19-oic acid had the highest apparent growth-inhibitory activity against *S. epidermidis* (MIC = 32 μg/mL), although it displayed only low inhibitory activity against *S. aureus*. In contrast, the flavonoid compound jaceosidine displayed the highest growth-inhibitory activity against *S. aureus* (MIC = 128 μg/mL). 

Decoctions prepared from *Eremophila serrulata* leaves were also traditionally used to treat bacterial infections (Appendix A). In one study, *E. serrulata* leaves were extracted with diethyl ether, and then the extract was fractionated using RP-HPLC [18]. Several compounds with antibacterial activity were isolated and identified. The compounds 9-methyl-3-(4-methyl-3-pentenyl)-2,3-dihydronaphtho[1,8-bc] pyran-7,8-dione and 8,20-diacetoxyserrulat-14-en-19-oic acid displayed particularly good antimicrobial activity against *Staphylococcus aureus* (ATCC 29213), with MIC values ranging from 15.6 to 250 μg/mL. Both of these compounds also inhibited the growth of other Gram-positive bacteria, including *Streptococcus pyogenes* and *Streptococcus pneumonia*, although they were ineffective against all of the Gram-negative bacteria species tested.

Another study by the same group screened two surrulatane diterpenoids isolated from *Eremophila neglecta* to determine the growth-inhibitory activity against an extended panel of human bacterial pathogens (including several MRSA strains) as well as multiple Gram-positive and Gram-negative species using broth dilution assays [19]. The serrulatan compounds 8,19- dihydroxyserrulat-14-ene and 8-hydroxyserrulat-14-en-19-oic acid were both effective inhibitors of all of the Gram-positive bacteria tested, with MIC values between 3 and 165 μM and MBC values between 6 and 330 μM. Notably, both compounds had noteworthy anti-mycobacterial activity against *Mycobacterium fortuitum* and *Mycobacterium chelonae*. In contrast, *Moraxella catarrhalis* was the only Gram-negative bacterium of the panel tested that was inhibited by the serrulatane diterpenoids with MICs and MBCs of 3–20 μM. The lack of activity against most of the Gram-negative bacteria may be related to their decreased passage through the cell wall of Gram-negative bacteria. It is likely that the relatively large molecular size and bicyclic conformation of serrulatans may inhibit their entry into the size-selective porin channels of the outer membrane of the Gram-negative cell wall, thereby preventing the compounds from reaching the inner membrane and cytoplasm of Gram-negative bacterial species [20]. Of further interest, a different study also reported that 8-hydroxyserrulat-14-en-19-oic acid isolated from *E. neglecta* leaves inhibits the formation of *S. epidermidis* and *S. aureus* biofilms and also has a dispersing effect on established biofilms [21].

Essential oils prepared by the hydro-distillation of *Eremophila longifolia* leaves were tested using both disc diffusion and broth dilution assays [22]. The authors detected antibacterial components using bioautography and reported an interesting trend. Essential oils rich in monoterpenes and hydrocarbon monoterpenols had moderate antimicrobial activity against a panel of Gram-positive and Gram-negative bacteria, including *S. aureus* and *S. epidermidis*, whereas essential oils that contained ketone compounds as the dominant components displayed substantially lower antibacterial activity. Another study by the same group tested *E. bignoniiflora* leaf essential oil for antibacterial activity against a panel of bacteria [23]. The oil showed moderate to high activity against all of the Gram-positive organisms, yet only low growth-inhibitory activity against the Gram-negative bacteria. Bioautography of the Gram-positive bacteria indicated that the antimicrobial activity was related to the higher-polarity components of the essential oil, instead of to fenchyl, bornyl, and nerol acetate, which were the major constituents of the essential oils.

### 3.2. Antifungal and Antiviral Properties

Multiple *Eremophila* spp. also have antifungal properties. Several studies examined the antifungal properties of *Eremophila* spp. in parallel with their antibacterial activity. Some of those studies have already been summarised in the preceding section. Additionally, a recent study screened *E. alternifolia* leaf extracts and isolated compounds for antiviral activity and reported substantial activity [13]. The authors tested the extract and isolated compounds using disk diffusion and broth microdilution assays against ten clinically relevant yeast and mould species. The most potent activity was observed for the diterpene compound 8,19-dihydroxyserrulat-14-ene against *Cryptococcus gattii* and *Cryptococcus neoformans*, with MICs comparable to those of amphotericin B. This compound was similarly active against six *Candida* species (including *C. albicans, C. krusei*, and *C. glabrata*) and is therefore a promising drug target for the development of a novel therapy to treat infections of *Cryptococcus* spp. and other yeast species. The authors of that study reported that 8,19-dihydroxyserrulat-14-ene inhibits several biosynthetic pathways and compromises cell membrane integrity, thereby resulting in cell death.

A further study reported that essential oils produced by hydro-distillation of *Eremophila longifolia* leaves had good inhibitory activity against the human dermal pathogens *Trichophyton interdigitale, Trichophyton rubrum*, and *Trichophyton mentagrophytes* [22]. The authors also evaluated the phytochemistry of the essential oils and reported that the antifungal activity was related to the borneol content of the oils. Interestingly, the authors also noted substantially greater antifungal activity when essential oils were prepared using leaves that were burned or partially pyrolysed during hydro-distillation. This is a particularly interesting finding and may provide some explanation as to why the First Australians frequently burned the leaves of this species in smoking ceremonies. The authors of that study reported that monoterpenol-dominated oils had moderate antifungal activity, although this activity increased significantly when the oils were partially pyrolysed. The *E. longifolia* essential oils were also effective against *Candida albicans*, possibly indicating that this species may be a good general fungicide. A similar study by the same group also tested an essential oil prepared from *Eremophila bignoniiflora* leaves using broth dilution assays against the same *Trichophyton* species, with moderate activity reported against all dermatophytes and against *C. albicans* [23].

Several *Eremophila* spp. also have antiviral bioactivities. Indeed, a study was conducted using *Eremophila latrobei* subsp. glabra extracts to examine their antiviral activity [24]. Notably, *Eremophila latrobei* subsp. *glabra* leaf extracts displayed antiviral activity against Ross River virus (RRV) at non-cytotoxic concentrations, indicating its potential for therapeutic use. The study was conducted on 40 plant species used traditionally by First Australians to treat viral diseases. However, that study did not determine the antiviral components, nor was the antiviral mechanism determined. Substantially more work is required to thoroughly evaluate the potential of this species to treat Ross River fever. Additionally, the extracts should also be evaluated for antiviral activity against other viral pathogens, particularly against other RNA viruses.

### 3.3. Other Therapeutic Properties

Multiple other therapeutic effects have also been described for *Eremophila* spp. preparations, including anti-inflammatory effects and immunomodulatory effects. Extracts prepared using *Eremophila sturtii* leaves have substantial anti-inflammatory activity [16]. Indeed, an *E. sturtii* ethanolic leaf extract inhibited cyclooxygenase 1 (COX-1) and cyclooxygenase 2 (COX-2) by 95% and 89%, respectively, at 2 mg/mL. In contrast, the extract was completely ineffective as an inhibitor of 5-lipoxygenase (5-LOX). Furthermore, hexane, ethyl acetate, n-butanol, and water fractions were prepared from the extract and evaluated for COX-1, COX-2, and 5-LOX activity, with varying potencies being reported. The ethyl acetate fraction was the most active, achieving 88% and 66% inhibition of COX-1 and COX-2, respectively, at 2 mg/mL. Two serrulatane diterpenoids were subsequently isolated from the ethyl acetate fraction and were identified as 3,8-dihydroxyserrulatic acid and serrulatic acid. Additionally, the triterpenoid β-sitosterol was also identified in the fraction. Isolated 3,8-dihydroxyserrulatic acid had weak COX-1 and COX-2 inhibitory activity, inhibiting enzyme activity by 57% and 14%, respectively, at 1 mg/mL. In contrast, serrulatic acid had substantially greater inhibitory activity against COX-1 and COX-2 at 1 mg/mL, with 99% and 97% inhibition, respectively. The efficacy of that compound was further quantified, and IC_50_ values of 27 and 73 μg/mL were determined for COX-1 and COX-2 inhibition, respectively. In comparison, IC_50_ values of 0.1 and 0.6 μg/mL were determined for ibuprofen against COX-1 and COX-2, respectively.

A more recent study evaluated an *Eremophila neglata* leaf extract for COX-1 and COX-2 inhibitory activity and reported similar enzyme inhibitory [21]. That study also isolated 8-hydroxyserrulat-14-en-19-oic acid from *E. neglata* leaves and evaluated its anti-inflammatory potential against different inflammatory biomarkers. That compound displayed substantial inhibitory effects on tumour necrosis factor alpha (TNF-α) and interleukin-6 (IL-6) secretion from bone marrow-derived macrophage (BMDM) cells, although interleukin-1β (IL-1β) secretion was not significantly altered. Another study examined *Eremophila neglecta* leaf extracts and serrulatan diterpenoid compounds isolated from them and reported substantial cytotoxic effects against Vero (monkey kidney cells), with similar cytotoxicity levels evaluated for 8,19-dihydroxyserrulat-14-ene and 8-hydroxyserrulat-14-en-19-oic acid [19]. Notably, the cytotoxicity of these compounds may impact their therapeutic potential, and substantially more work is required to evaluate their cytotoxicity and therapeutic index before they can be safely used clinically.

Several studies have also screened an *Eremophila glabra* leaf extract against some diabetic biomarker enzymes [25]. The extract showed good inhibitory activity against α-glucosidase and protein tyrosine phosphatase 1B (PTP1B) enzymes, with IC_50_ values of 19.3 μg/mL and 11.8 μg/mL, respectively. The authors also used α-glucosidase/PTP1B inhibition profiling combined with HPLC-PDA-HRMS and NMR to isolate and identify compounds contributing to the inhibitory activities of these enzymes. The extract was rich in serrulatan diterpenoids. Two out of seven isolated serrulatan diterpenoids showed weak PTP1B inhibitory activity, and one showed dual α-glucosidase and PTP1B inhibitory activity, indicating that *E. glabra* leaves are promising drug targets for antidiabetic drug discovery, although substantially more work is required in this area. Another study tested an *Eremophila bignoniiflora* leaf extract against protein tyrosine phosphatase 1B and reported an IC_50_ of 23.9 µg/mL [26], indicating that it also has potential to treat diabetes.

Some *Eremohilia* species may also have cardioactive effects. One study reported that the glycoside verbascoside, which was isolated from the methanolic and aqueous extracts of *E. alternifolia* leaf extracts, induced significant increases in chronotropism, inotropism, and coronary perfusion rates (CPR) in Langendorffrat hearts [27]. Another study designed to evaluate the antioxidant and hepatoprotective activities of a methanolic extract of *E. maculata* leaf extract in vitro and in a rat in vivo model reported promising antioxidant and hepatoprotective activities [28]. The same study also performed a detailed phytochemical study on the extract, followed by molecular docking experiments on TNF-α to verify the efficacy of the isolated compounds. The authors concluded that the antioxidant and hepatoprotective activities are due (at least in part) to *E. maculata* lignans and phenylpropanoids.

## 4. Phytochemistry and Antioxidant Content

To date, more than 200 secondary metabolites from multiple phytochemical categories have been identified in plants of the genus *Eremophila*. The major classes of secondary metabolites identified include monoterpenoids, sesquiterpenoids, diterpenes (including eremane, cembrene, decipiane, serrulatane, and viscidane class diterpenoids), triterpenoids, verbascosides, cyanogenic glucosides, fatty acids, phenylpropanoids, lignans, and flavonoids [10].

### 4.1. Antioxidant Capacity

Cellular oxidation reactions may result in the formation of molecular entities that contain one or more unpaired (and therefore unstable) electrons, which confer these “free radicals” with increased reactivity. Whilst the production of free radicals is a normal result of cellular respiration, their production is increased by several biological stimuli, including inflammation and stress, as well as immune responses [29]. Additionally, free radical production is also increased in response to some environmental stimuli, including smoking, pollution, and prolonged exposure to the sun. Free radicals may comprise reactive oxygen species (ROS) and reactive nitrogen species (RNS), both of which contribute to normal immune function and cellular control at normal levels. However, external stresses can create an imbalance in pro-oxidant/antioxidant concentrations that may induce oxidative stress, which can induce several chronic and degenerative diseases. For example, uncontrolled oxidative stress can cause failures in cellular repair mechanisms, which may result in cancer generation [30,31]. Additionally, oxidative stress can cause damage to guanine nucleotides and may subsequently result in Parkinson’s disease [30,32]. Oxidative stress may also induce serious diseases, including cancer, diabetes, rheumatoid arthritis, cardiovascular, and neurodegenerative diseases, and premature aging.

Antioxidants moderate the formation of free radicals and thereby block oxidative stress and its associated health problems. Cells utilise both endogenous enzymes (including catalase, glutathione peroxidase, superoxide dismutase, and thioredoxin) and non-enzymatic antioxidants (including vitamins A, C, E, flavonoids, and some dietary minerals, including selenium and zinc) to reduce oxidative stress [33]. Indeed, the potential of antioxidants to reduce oxidative stress-related diseases has been well reported. The increased intake of dietary flavonoids has been shown to reduce the incidence of some cancers [30]. Additionally, the increased intake of dietary antioxidants normalises intra- and extracellular pH levels in patients suffering from Alzheimer’s disease, thereby blocking the oxidative stress-related cellular damage apparent in that disease [34]. Similarly, non-enzymatic antioxidants are effective in the management of diabetes, hypertension, and hyperlipidaemia [35]. Plant-based foods are the main dietary sources of antioxidants, with spinach, carrots, kiwi fruit, citrus fruits, red fruits, apples, cabbage, etc., having particular oxygen radical absorbance capacities (ORAC) and total antioxidant capacities (TAC). Additionally, other foods including chocolate, red wine, tea, shellfish, and some spices also have a high ORC and TAC values and are therefore considered beneficial to include in one’s diet.

### 4.2. Flavonoids and Phenolic Compounds

Flavonoids and phenolic compounds are the main antioxidant phytochemical molecules found in plants [36,37]. These compounds are synthesised by plants to promote growth and to protect the plant in adverse conditions [38]. In addition to their noteworthy antioxidant activities, multiple other therapeutic effects have been documented for flavonoids, including antimicrobial, anti-proliferative/anticancer, anti-inflammatory, antidiabetic, and anti-mutagenic properties. Notably, the consumption of foods rich in flavonoids decreases the incidences of some chronic diseases, including (but not limited to) Alzheimer’s and other neurodegenerative disorders as well as cardiovascular diseases [39,40]. Flavonoids inhibit the activity of the enzymes xanthine oxidase, cyclooxygenase, lipoxygenase, and phosphoinositide-3-kinase, thereby modulating various chronic diseases [39]. Therefore, high flavonoid contents in plant preparations indicate that the preparations may have noteworthy therapeutic properties and may highlight the need to test that species.

Previous phytochemical analyses of *Eremophila* spp. have identified multiple flavonoids and phenolic compounds with substantial antioxidant capacity. Notably, many of the *Eremophila* spp. that have been determined to be rich in flavonoids have been reported to exhibit antiprotozoal, antioxidant, antibacterial, and cytotoxic properties [10], further verifying their therapeutic potential. In particular, the flavonoids pinobanksin (Figure 2a) and galangin-3-methyl ether (Figure 2b) have been isolated, and high levels of pinobanksin in extracts prepared from *Eremophilia alternifolia* R. Br. Leaves have been identified [41]. A more recent study also isolated mg quantities of pinobanksin from approximately 100 g of fresh *E. alternifolia* leaves and confirmed the its identity and the identity of similar yields of two further flavonoids: pinobanksin-3-cinnamate (Figure 2c) and pinobanksin-3-acetate (Figure 2d) [11]. Interestingly, that study also reported substantial antibacterial activity for pinobanksin-3-cinnamate (MIC = 1020 mM) against several strains of the Gram-positive bacterium *Staphylococcus aureus*. Of particular interest, the authors of that study reported that pinobanksin-3-cinnamate also had noteworthy inhibitory activity against methicillin-resistant (MRSA) and biofilm-forming strains of *S. aureus*. In contrast, pinobanksin-3-cinnamate (and the other flavonoids isolated in that study) were completely ineffective against the Gram-negative bacterium *Escherichia coli*. However, that study screened the compounds against a limited bacterial panel, and it is unclear whether the difference in efficacy between these bacteria is due to an inability to cross the outer lipid layer of Gram-negative bacteria, or if it is specific to the strain of *E. coli* tested. Further studies are needed to test these flavonoids against extended panels of both Gram-positive and Gram-negative bacteria and to determine the antibacterial mechanism.

Three further flavonoids have been identified as major components (0.01–0.05% of the original plant material mass) of *Eremophila glabra* (R.Br.) Ostenf. Leaves: hispidulin (Figure 2e), jaceosidin (Figure 2f), and cirsimaritin (Figure 2g) [17]. Notably, that study also reported good antibacterial activity for the isolated flavonoids, with MICs below 512 μg/mL recorded against multiple bacteria. The inhibitory activity of jaceosidin was particularly noteworthy, with an MIC of 128 μg/mL measured against a reference *S. aureus* strain (NCTC10442). A similar MIC value was also determined for hispidulin in that study. Notably, jaceosidin has also been identified in other *Eremophilia* spp., including *Eremophila microtheca* (F.Muell. ex Benth.) F.Muell., with the dry leaves having a yield of ~0.1% (m/m) [42]. That study reported good activity (MICs 16–32 μg/mL) for jaceosidin against several *S. aureus* isolates. Interestingly, the potency of jaceosidin reported against the *S. aureus* strains tested in that study was substantially greater than that reported against the reference strain in the Algreibya et al. (2018) study [17]. It would be interesting to compare the resistance profiles of all of these strains in future studies and to determine the antibacterial mechanism of this compound. An additional flavonoid, 3′,5,5′-trihydroxy3,4′,6,7-tetramethoxyflavone (Figure 2h), was also identified in *Eremophila fraseri* F.Muell., although the yields of this compound were not specified in those studies [41,43,44].

A recent study isolated flavonoids (yields not specified) and identified an additional eleven (as well as numerous terpenoids and other compounds) in an *Eremophilia bignoniiflora* (Benth.) F.Muell. leaf extract [26]. That study used liquid chromatography, high-resolution mass spectroscopy (LC-HRMS), and nuclear magnetic resonance spectroscopy (NMR) to identify taxifolin (Figure 2i), dihydrokaempferol (Figure 2j), 3-O-acetyltaxifolin (Figure 2k), quercetin-3-methyl ester (Figure 2l), pinobanksin (Figure 2a), 3-acetoxyhespertin (Figure 2m), pinocembrin (Figure 2n), galangin (Figure 2o), pinobanksin-3-acetate (Figure 2p), and galangin-3-methyl ether (Figure 2q). That study also reported that all of these flavonoids had substantial protein tyrosine phosthatase 1B (PTP1B) inhibitory activity, with IC_50_ values as low as 24 μg/mL. As PTP1B has been shown to regulate multiple pathogenic mechanisms for Alzheimer’s disease, these compounds (and crude *E. bignoniiflora* preparations) have the potential to treat that disease. PTP1B is also implicated in diabetes mellitus pathogenesis, and these flavonoids therefore also have potential to treat that disease. The same study also tested crude the *E. bignoniiflora* extract (although not the isolated flavonoids) for its ability to inhibit other enzymes associated with diabetes (α-glucosidase, α-amylase) [26]. Substantial inhibition was noted, although IC_50_ values were not reported, as the extract did not inhibit enzyme activity by >50% at any of the concentrations tested. Further studies are required to quantify this activity and to determine whether the flavonoid compounds contribute to this activity.

### 4.3. Terpenoids

Terpenoids are a major class of phytochemicals present in *Eremophila* spp. and are likely to contribute to many of their therapeutic properties (particularly antimicrobial properties). Terpenoids are produced by a myriad of higher plants, some mosses and liverworts, as well as by some insects, microbes, and marine organisms. In plants, they are produced as protective molecules, and their anti-infective properties have been well documented [45,46,47,48,49,50]. Additionally, several studies have reported therapeutic effects for the terpenoid components of several *Eremophila* spp. [51].

Numerous monoterpenoids have been identified in *Eremophila* spp. as major components. In particular, the monoterpenoid D-limonene (Figure 3a) has been reported in the essential oils prepared from leaves of several individual *E. longifolia* trees [22] and *E. alternifolia* in 0.2–40% relative abundances [52]. Additionally, the monoterpenoid borneol (Figure 3b) was also detected in *E. longifolia* leaves, although its levels varied widely between essential oils prepared from different cultivars (from below detection thresholds to ~35% relative abundance) [22]. Notably, that study also reported substantial inhibitory activity for the isolated borneol against the fungal skin pathogens *Trichophyton interdigitalis, Trichophyton rubrum*, and *Trichophyton mentagrophytes* using agar transplantation methods, indicating that it may be useful for treating ringworm, tinea, and other fungal skin infections. Interestingly, that study reported that the anti-*Trichophyton* activity was the highest when leaves that were accidently burnt or partially pyrolysed during hydro-distillation were used to produce the essential oils. Additionally, that study also identified α-pinene (Figure 3c) in *E. longifolia* essential oils, which was also the highest in essential oils produced using partially burnt leaves (up to 19% relative abundance in partially pyrolised leaves compared to 0.01–4.3% in fresh leaves) [40]. Notably, α-pinene is a common component of multiple *Eremophila* spp., and it has also been identified in similar relative abundances in essential oils prepared from fresh *E. freelingii, E. longifolia, E. duttonii*, and *E. sturtii* leaves [8].

The monoterpenoid composition of *E. longifolia* has been particularly well reported. In addition to the compounds outlined above, a number of other monoterpenoids have also been identified in *E. longifolia* essential oil. A previous study that used bioautography to examine the antibacterial activity of *E. longifolia* components reported that menthone (Figure 3d) was the most potent antibacterial component in the essential oil [22]. However, that study reported only low relative abundances (≤0.2%) for methone in those preparations. Additionally, another study also identified and noted antibacterial activity for terpinolene (Figure 3e), α-terpineol (Figure 3f), and sabinene (Figure 3g) in relative abundances ranging from ~0.1 to 19%. An additional monoterpenoid, fenchone (Figure 3h), was identified in *E. alternifolia* essential oils prepared from the leaves in similar abundances [52]. Additionally, the modified monoterpenoids fenchyl-acetate (Figure 3i) and bornyl-acetate (Figure 3j) were identified in the hydrodistillate prepared from *E. bignoniiflora* leaves (0.5–1.8% relative abundances) [23]. Notably, that study also assessed the antioxidant activity of these components and reported significant (but moderate) free radical scavenging activity. It is possible that this activity may contribute to the therapeutic potential of *E. alternifolia* essential oils and extracts. Another study identified cineole (Figure 3k) in *E. dalyana* essential oil, although the yields of this compound were not specified [53].

Similar to several other *Eremophila* spp. discussed above, D-limonene and sabinene were also detected in *E. maculata* essential oil, although the yields were not specified [54]. Notably, sabinene has noteworthy inhibitory activity against several bacterial strains as well as against fungi and yeast [54]. The same studies reported that D-limonene inhibits the growth of multiple bacteria (both Gram-positive and Gram-negative species) and of the yeast *Candida albicans*. Subsequent studies have determined that exposure to limonene can elicit changes in microbial cell wall structure and function by modulating the expression of the ROM1 and RLM1 genes [55].

*Eremophilia* spp. are also rich sources of sesquiterpenoids, and multiple compounds have been identified. Many of these compounds are common across multiple *Eremophilia* spp. For example, the sesquiterpenoids elemol (Figure 4a) and eudesmol (Figure 4b) have been identified in both *E. dalyana* [53] and *E. cuneifolia* essential oils, although the yields were not specified in those studies [51]. Another study isolated emol and eudesmol and tested them against a microbial panel [56]. Interestingly, that study reported strong antimicrobial activity for both sesquiterpenoids against multiple pathogens. The related sesquiterpenoid eremophilone (Figure 4c) and several related structural analogues have also been identified in *E. mitchellii* leaf essential oil as major components (relative abundances were not specified) [57,58,59,60]. Interestingly, several eremophilones are cytotoxic towards P388D1 mouse lymphoblasts, with IC_50_ values between 42 and 110 mg/mL [61], indicating that these compounds may be useful against some cancers, Additionally, another study also identified spathulenol (Figure 4d) and 10,11dehydrongaione (Figure 4e) in *E. cuneifolia* essential oil in unspecified relative abundances [51]. Spathulenol was also identified in *E. paisley* leaf resin, although the yield was not reported [52]. Another study reported that 10, 11-dehydrongaione has moderate antibacterial activity against several bacteria [20].

Furanoid sesquiterpenoids are also relatively common and have been reported in several *Eremophila* spp. The furanoid sesquiterpenes 9-hydroxydihydromyomontanone (Figure 4f) and 4-hydroxydihydromyodesmone (Figure 4g) have been isolated from *E. alternifolia* leaf extracts along with several β-ketol derivatives (yields were not specified) [62]. Similarly, freelingyne (Figure 4h) and several analogues (unspecified yields) have been identified in *E. freelingii* leaf essential oil [60,63,64]. The furanoid sesquiterpene genifuranal (Figure 4i) was identified in *E. longifolia* [3]. That study reported genifuranal as a major component of the smoke extract, although the authors did not specify the yield of this compound. Interestingly, genifuranal was only detected in appreciable quantities in that study in the smoke collected from burning *E. longifolia* leaves. None was detected when the leaves were not burned. This is interesting, as *E. longifolia* were often used by the First Australians in smoking ceremonies, partially validating the usage method. Additionally, five previously unreported 2(5H)-furanone sesquiterpenes were recently identified in *E. bignoniiflora* leaves as major extract components (yields were not specified) [65]. Notably, the authors reported that the furanone sesquiterpenes exhibited moderate PTP1B inhibitory activity in vitro. As PTP1B modulates multiple processes in both Alzheimer’s disease and diabetes mellitus [65], 2(5H)-furanone has potential for the treatment of these conditions, although its effects are yet to be confirmed.

Recently, seven previously unreported caryophyllane sesquiterpenoids (Figure 4j–p) were isolated from *E. spathulata* leaves and were structurally identified, although the authors did not provide the yields of the individual components [66]. However, as this is the first report of these compounds in this species, these compounds may be present in *E. spathulata* leaves in relatively low abundances, although this remains to be verified. Interestingly, structurally similar caryophyllanes have been reported to have multiple therapeutic activities, including anti-inflammatory, antibacterial, antifungal, and chemosensitizing properties [67]. However, the *E. spathulata* leaf caryophyllane sesquiterpenoids are yet to be checked for similar activities, and this remains to be confirmed.

A wide variety of diterpenoids have also been identified in *Eremophila* spp. The serrulatane diterpenoids are represented in multiple species. A recent study purified seven serrulatane diterpenes from *E. glabra* leaf essential oils, with extraction yields ranging from 0.01 to 0.05% (m/m) of the dry leaf material [17]. Several of those terpenoids were identified, including 18-acetoxy-8,20-dihydroxyserrulat-14-en-19-oic acid (Figure 5a), 18,20-diacetoxy-8-hydroxyserrulat-14-en-19-oic acid (Figure 5b), 8,18,20-triacetoxyserrulat-14-en-19-oic acid (Figure 5c), 18-acetoxy-8-hydroxyserrulat-14-en-19-oic acid (Figure 5d), 8,20-diacetoxyserrulat-14-en-19-oic acid (Figure 5e), 8,18,20-trihydroxyserrulat-14- en-19-oic acid (Figure 5f), and 20-acetoxy-8-hydroxyserrulat-14-en-19-oic acid (Figure 5g). The authors of that study screened the isolated serrulatane diterpenoids for antibacterial activity against a panel of Gram-positive pathogens and reported noteworthy activity for 8,18,20-triacetoxyserrulat-14-en-19-oic acid, 18-acetoxy-8-hydroxyserrulat-14-en-19-oic acid, and 20-acetoxy-8-hydroxyserrulat-14-en-19-oic acid, with MIC values ranging from 32 to 512 μg/mL. Notably, 18-acetoxy-8-hydroxyserrulat-14-en-19-oic acid has also been identified in several other *Eremophilia* spp., including *E. alternifolia*, with extraction yields of 0.08–1.25% (m/m) being obtained from the dried leaf material [12].

18-Acetoxy-8-hydroxyserrulat-14-en-19-oic acid and four additional serrulatane diterpenoids have been isolated from *E. glabra*: 8,16-dihydroxyserrulat-19-oic acid (Figure 5h): 8-hydroxy-16-[4-methylpent-3-enoyloxy]serrulat-19-oic acid (Figure 5i), 8-hydroxy-16-hydrocinnamoyloxyserrulat-19-oic acid (Figure 5j), 8-hydroxy-16-cinnamoyloxyserrulat-19-oic acid (Figure 5k), and 8-hydroxyserrulat-14- en-19-oic acid (Figure 5l), with yields of 0.01–0.05% (m/m) being obtained the dried leaves [17]. Multiple serrulatane diterpenoids have also been identified in *E. duttonii* leaves, including 8, 20-trihydroxyserrulat-14-ene (Figure 5m, and serrulat-14-en-3, 7, 8, 20-tetraol (Figure 5n), although the extraction yields were not specified in those studies [20,67,68,69]. One of those studies also reported that both of these compounds had noteworthy antibacterial activity towards *Staphylococcus aureus* [20].

Another three additional serrulatane diterpenoids were identified in *E. microtheca*: 3-acetoxy-7,8-dihydroxyserrulat-14-en-19-oic acid (Figure 5o), 3,7,8-trihydroxyserrulat-14-en-19-oic acid (Figure 5p), and 3,19-diacetoxy-8-hydroxyserrulat-14-ene (Figure 5q), with yields for the individual diterpenoids ranging from 0.4 to 4.4% of the mass of the dried *E. microtheca* leaves [42]. Additionally, acetylation and methylation of the *E. microtheca* serrulatane diterpenoids resulted in the formation of 3, 8-diacetoxy-7-hydroxyserrulat-14-en-19-oic acid (Figure 5r) and 3, 7, 8-trihydroxyserrulat-14-en-19-oic acid methyl ester (Figure 5s), respectively. That study screened all of these terpenoids for antibacterial activity against a panel of Gram-positive and Gram-negative bacteria and reported that all of the compounds exhibited moderate activity against *Streptococcus pyogenes*, with MIC values in the range of 64–128 μg/mL. In contrast, *Enterococcus faecalis* and *Enterococcus faecium* were unaffected by all of the *E. microtheca* serrulatane diterpenoids.

Similar serrulatane diterpenoids as those already described have also been isolated from other *Eremophila* spp. As well as being identified in *E. glabra* leaf essential oil, 8, 20-dihydroxyserrulat- 14en-19-oic acid was also identified in *E. paisley* leaves [52]. Similarly, 3, 8-dihydroxyserrulatic acid was identified in *E. sturtii* leaf extracts, although relative abundances were not specified in either of those studies [34]. The *E. sturtii* study also reported that 3, 8-dihydroxyserrulatic acid inhibited the growth of the Gram-positive bacterium *Staphylococcus aureus* and the Gram-negative bacteria *Escherichia coli* and *Pseudomonas aeruginosa* [34]. Additionally, the diterpenoid also inhibited the growth of the yeast *Candida albicans* and inhibited the inflammatory cyclooxygenase enzymes COX-1 and COX-2 as well as 5- lipoxygenase (5-LOX). Thus, this compound has potential as both an antibacterial and an anti-inflammatory therapy. Additionally, 8-hydroxyserrulat-14-en-19-oic acid (which is also present in *E. glabra*) was isolated from *E. neglecta* (yield was not specified) [19]. The authors of that study also reported that this compound inhibited the growth of a panel of human and veterinary bacterial pathogens, including some multidrug-resistant strains. Of particular note, this diterpenoid inhibited the growth of *Mycobacterium fortuitum, Mycobacterium chelonae*, and *Moraxella catarrhalis*. Additionally, 8, 20-trihydroxyserrulat-14-ene was isolated from *E. phyllopoda* leaves and branches (yield was not specified) [70]. However, that study also identified a novel serrulatane diterpenoid (Figure 5t).

Different classes of diterpenoids have also been identified in *Eremophila* spp. The viscidane-type diterpenoid iridoid geniposidic acid (Figure 6a) and the eremane type diterpenoid 5-béta-hydroxy- 16-oxoereman- 19-oic acid (Figure 6b) have been isolated from *E. cuneifolia* (yields were not specified) [51]. Both of those diterpenoids have also been identified in *E. gilesii* leaves (unspecified yields) [52,71]. Additionally, the cembranoid-malonyl ester 3,15- epoxy- 19-oxocembra-7,11 -dien- 18-ol (Figure 6c) has also been identified in *E. gilesii* and *E. fraseri*, although the yields were not specified in those studies [41,43,44,72].

### 4.4. Other Classes of Eremophilla spp. Phytochemicals

Several iridoid glycosides have also been identified in *Eremophila* spp. leaves from several species. Geniposidic acid (Figure 6d) has been isolated from *E. cuneifolia* leaves [51] as well as from *E. longifolia* [3] and *E. alternifolia* (extraction yields were not specified) [27,73]. Similarly, the caffeoyl phenylethanoid glycoside verbascoside (Figure 6e) has been reported in *E. cuneifolia* leaves [71], *E. longifolia* leaves [3], *E. gilesii* aerial parts [74], *E. microtheca* leaves [42], and *E. glabra* leaves, with yields of ~0.01% of the mass of the dried plant material in each case [17]. Notably, several studies have examined the therapeutic potential of verbascoside and have reported that it has significant cardioactive effects [27,73]. Additionally, it inhibits ADP-induced human platelet aggregation and serotonin release [74].

Several lignans have also been reported in *Eremepholia* spp., including epieudesmin (Figure 6f), which has been isolated from *E. dalyana* leaf extracts, although the yield was not specified in that study [53]. Additionally, pinoresinol-4-Oβ-D-glucopyranoside (Figure 6g) has been isolated from *E. maculata* leaves and has been shown to be a strong TNF-α inhibitor [28]. It also has noteworthy antimicrobial and xanthine oxidase inhibitory activity [10,48]. Unfortunately, that study did not specify the extraction yield of pinoresinol-4-Oβ-D-glucopyranoside from *E. dalyana* leaves, and further studies are required to quantify the levels of this compound. Three further lignans isolated from *E. phyllopoda* leaves were identified as sesamin (Figure 6h), episesamin (Figure 6i), and aptosimon, but their levels were not quantified in the extract (Figure 6j) [70].

## 5. Conclusions

This review of the traditional medicinal properties, phytochemistry, and pharmacognosy of plants of the genus *Eremophila* has highlighted the therapeutic importance of this useful genus of plants. *Eremophila* species have been used in First Australian traditional therapeutic systems in all areas in which they grow. However, despite the relative wealth of ethnopharmacological knowledge about the traditional usage of *Eremophila* species, rigorous scientific research has mainly focussed on a relatively few species, with the other species receiving substantially less attention. Furthermore, the work of relatively few research groups (particularly the Semple group in South Australia) dominate the field. In those species that have been most extensively studied, multiple therapeutic bioactivities have been reported, although examination of the antibacterial properties of *Eremophila* spp. has been the most extensively reported. Additionally, the bioactive phytochemical constituents have already been established for some of those species. Serrulatan diterpenoids appear to be ubiquitous across the genus and have been linked with the therapeutic properties of many species. However, for most species, the identification and mechanisms of action of the active principles have only been partially characterised, and substantially more work is required. Despite these previous studies, most of the *Eremophila* species screened to date have only been tested for one or two therapeutic properties, and screening against other bioactivities is required. At present, the testing of *Eremophila* species for therapeutic properties has focussed on the species with a documented history of use in traditional medicinal systems. However, the taxonomic relationship between *Eremophila* species suggests that testing other species from this genus for the same therapeutic properties that the more well-known species are used for is also warranted.

## Figures and Tables

**Figure 1 molecules-27-07734-f001:**
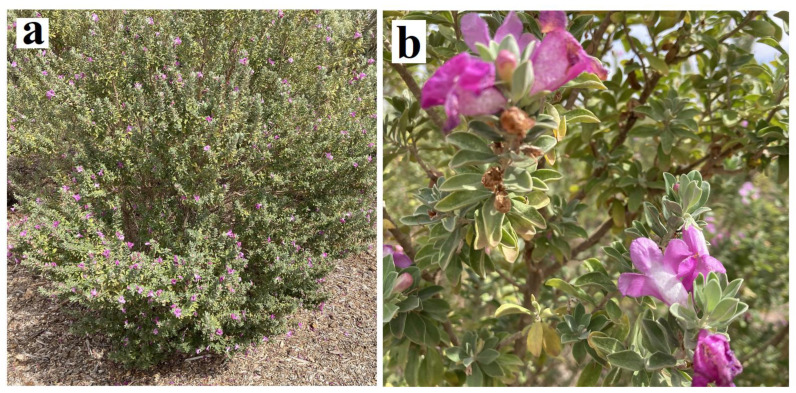
*Eremophila maculata* (Ker Gawl.) F.Muell. (**a**) whole plant and (**b**) leaves and flowers. Photograph was taken in the Australian Arid Lands Botanic Garden, Port Augusta, Australia, in January 2021 by Ian Cock.

**Figure 2 molecules-27-07734-f002:**
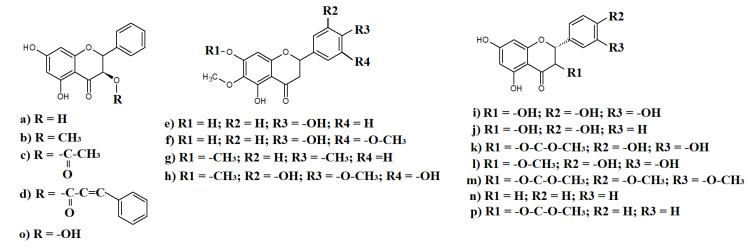
Some flavonoids detected in selected *Eremophila* spp.: (**a**) pinobanksin; (**b**) galangin-3-methyl ether; (**c**) pinobanksin-3-acetate; (**d**) pinobanksin-3-cinnamate; (**e**) hispidulin; (**f**) jaceosidin; (**g**) cirsimaritin; (**h**) 3′,5,5′-trihydroxy3,4′,6,7-tetramethoxyflavone; (**i**) taxifolin; (**j**) dihydrokaempferol; (**k**) 3-O-acetyltaxifolin; (**l**) quercetin-3-methyl ester; (**m**) 3-acetoxyhespertin; (**n**) pinocembrin; (**o**) galangin; (**p**) pinobanksin-3-acetate; and (**q**) galangin-3-methyl ether.

**Figure 3 molecules-27-07734-f003:**
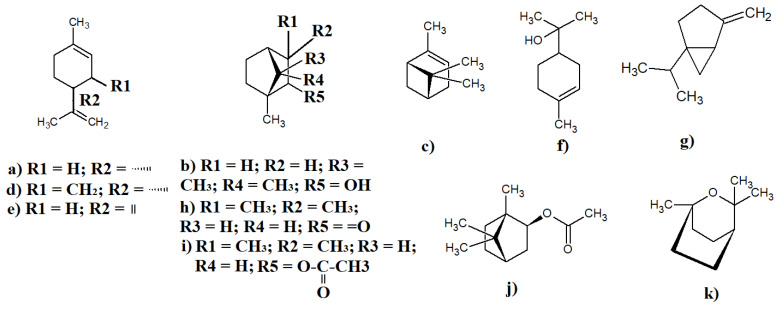
Selected monoterpenoids identified in *Eremophila* spp.: (**a**) D-limonene; (**b**) borneol; (**c**) α-pinene; (**d**) menthone; (**e**) terpinolene; (**f**) α-terpineol; (**g**) sabinene; (**h**) fenchone; (**i**) fenchyl-acetate; (**j**) bornyl-acetate; and (**k**) cineole.

**Figure 4 molecules-27-07734-f004:**
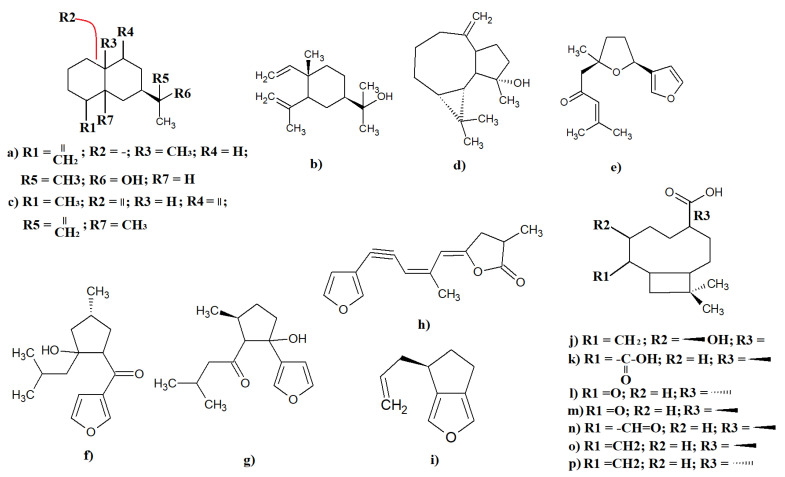
Selected sesquiterpenoids identified in *Eremophila* spp.: (**a**) elemol; (**b**) eudesmol; (**c**) eremophilone; (**d**) spathulenol; (**e**) 10,11-dehydrongaione; (**f**) 9-hydroxydihydromyomontanone; (**g**) 4- hydroxydihydromyodesmone; (**h**) freelingyne; (**i**) genifuranal; (**j**) (1,4,7,9)-7-hydroxy-11,11-dimethyl-8-methylenebicyclo[7.2.0]undecane-4-carboxylic acid; (**k**) (1,6,9)-10,10-dimethylbicyclo[7.2.0]undec-2-ene-2,6-dicarboxylic acid; (**l**) (1,4,9)-11,11-dimethyl-8-oxobicyclo[7.2.0]undecane-4-carboxylic acid; (**m**) (1,4,9)-11,11-dimethyl-8-oxobicyclo[7.2.0]undecane-4-carboxylic acid; (**n**) (1,4,9)-8-formyl-11,11-dimethylbicyclo[7.2.0]undec-7-ene-4-carboxylic acid; (**o**) (1,4,9)-11,11-dimethyl-8-methylenebicyclo[7.2.0]undecane-4-carboxylic acid; and (**p**) (1,4,9)-11,11-dimethyl-8-methylenebicyclo[7.2.0]undecane-4-carboxylic acid.

**Figure 5 molecules-27-07734-f005:**
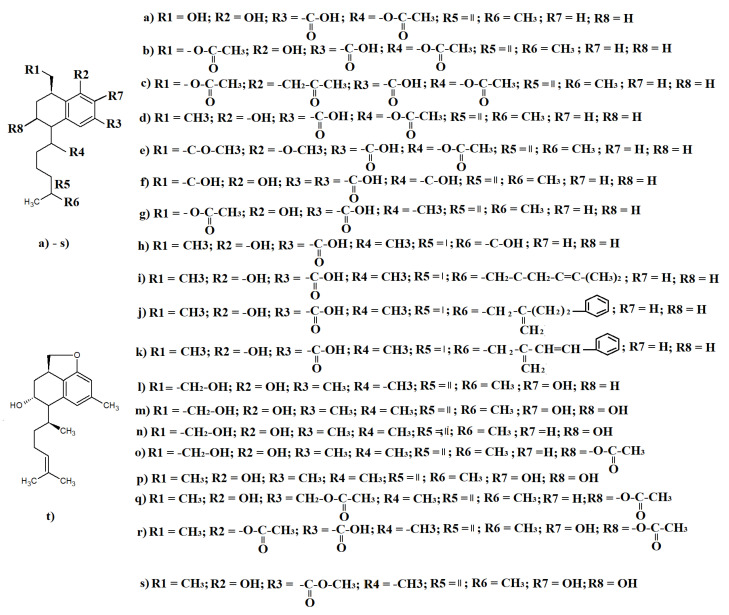
Selected serrulatane diterpenoids identified in *Eremophila* spp.: (**a**) 18-acetoxy-8,20-dihydroxyserrulat-14-en-19-oic acid; (**b**) 18,20-diacetoxy-8-hydroxyserrulat-14-en-19-oic acid; (**c**) 8,18,20-triacetoxyserrulat-14-en-19-oic acid; (**d**) 18-acetoxy-8-hydroxyserrulat-14-en-19-oic acid; (**e**) 8,20-diacetoxyserrulat-14-en-19-oic acid; (**f**) 8,18,20-trihydroxyserrulat-14- en-19-oic acid; (**g**) 20-acetoxy-8-hydroxyserrulat-14-en-19-oic acid; (**h**) 8,16-dihydroxyserrulat-19-oic acid; (**i**) 8-hydroxy-16-[4-methylpent-3-enoyloxy]serrulat-19-oic acid; (**j**) 8-hydroxy-16-hydrocinnamoyloxyserrulat-19-oic acid; (**k**) 8-hydroxy-16-cinnamoyloxyserrulat-19-oic acid; (**l**) 8-hydroxyserrulat-14- en-19-oic acid; (**m**) 8,20- trihydroxyserrulat-14-ene; (**n**) serrulat-14-en-3,7,8,20-tetraol; (**o**) 3-acetoxy-7,8-dihydroxyserrulat-14-en-19-oic acid; (**p**) 3,7,8-trihydroxyserrulat-14-en-19-oic acid; (**q**) 3,19-diacetoxy-8-hydroxyserrulat-14-ene; (**r**) 3,8-diacetoxy-7-hydroxyserrulat-14-en-19-oic acid; and (**s**) 3,7,8-trihydroxyserrulat-14-en-19-oic acid methyl ester.

**Figure 6 molecules-27-07734-f006:**
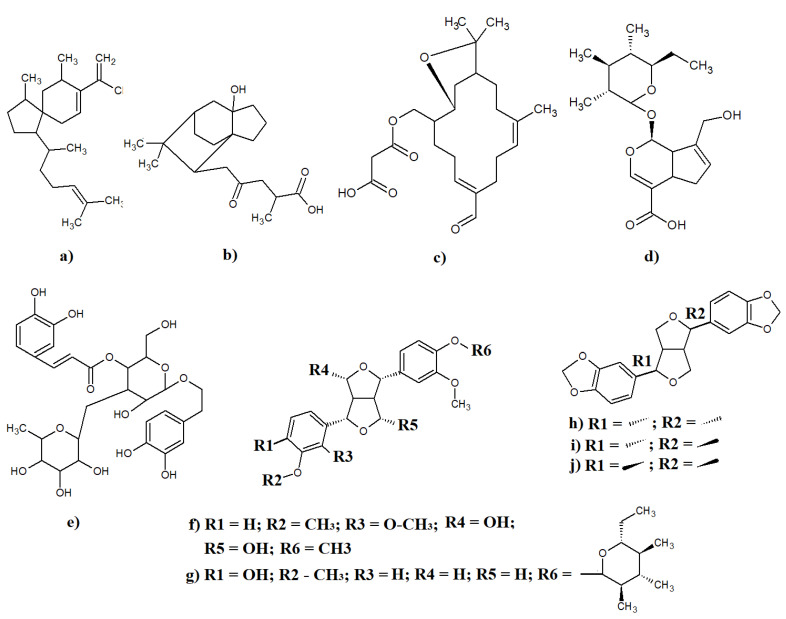
Non-serrulatane diterpenoids and other selected phytochemicals identified in *Eremophila* spp.: (**a**) 5 α-hydroxyviscida-3, 14-dien-20-oic acid; (**b**) -béta-hydroxy- 16-oxoereman- 19-oic acid; (**c**) 15- epoxy- 19-oxocembra-7,11 -dien- 18-ol; (**d**) geniposidic acid; (**e**) verbascoside; (**f**) epieudesmin; (**g**) pinoresinol-4-Oβ-D-glucopyranoside; (**h**) sesamin; (**i**) episesamin; and (**j**) aptosimon.

## Data Availability

Data are available from the corresponding author upon reasonable request.

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
