# Peer review of "Phytochemistry, Medicinal Properties, Bioactive Compounds, and Therapeutic Potential of the Genus Eremophila (Scrophulariaceae)"

_molecules, 2022, doi:10.3390/molecules27227734_

Round 1

Reviewer 1 Report

1) Since genus Eremophila belongs to the family Scrophulariaceae,  scientific classification demonstrating other genera in the same family should be shown.

2) For drawing chemical structures:

2.1) Some similar or related chemical structures drawn in Figs 2-6 should be condensed to save space and to easily understand in terms of structure-activity relationship. In addition, the compound names should be followed by bold running numbers (123...), for example pinobanksin (1). Thus, the bold running numbers should appear beneath each chemical structure.

2.2) Many functional groups in chemical structures are distorted, especially acetoxy group. The authors should consult organic chemists for proper drawing.

Author Response

  • Since genus Eremophila belongs tothe family Scrophulariaceae,  scientific classification demonstrating other genera in the same family should be shown.

The introduction section has been revised on lines 85-89 of the marked-up manuscript to address the reviewer’s comment:

“The taxonomy of the family Scrophulariaceae is complex and previously included approximately 5000 species from 275 genra. However, following reclassification, the family now consists of approximately 1830 species across 62 genra. A summary of the current phylogenetic classification of Eremophila within the family Scrophulariaceae is depicted in Supplementary Figure 1. The genus Eremophila (Tribe Myoporeae) is one of the largest genra of Scrophulariaceae, consisting of more than 260 species, all of which are endemic to Australia [4-6].”

Additionally, we have added the following figure to the Supplementary section:

Supplementary Figure 1. Classification of the genus Eremophila within the family Scrophulariaceae.

This figure has been included in the Supplementary section rather than to the main body of the manuscript. We believe that whilst this figure is of value for explaining the genus Eremophila and its relationship to other genra (as per the reviewer’s comment), it is peripheral to the focus of the study and therefore it is more suitable for the Supplementary data section.

2) For drawing chemical structures:

2.1) Some similar or related chemical structures drawn in Figs 2-6 should be condensed to save space and to easily understand in terms of structure-activity relationship. In addition, the compound names should be followed by bold running numbers (123...), for example pinobanksin (1). Thus, the bold running numbers should appear beneath each chemical structure.

The figures have been revised substantially to condense them as per the reviewer’s comment. However, instead of 75 individual numbered structure figures, we have kept the similar classes of compounds together within the individual figures for relevance and brevity of discussion. This has also allowed us to condense the figures as requested by the reviewer. Each structure within the individual figures is provided with a letter designation, which is provided in bold below the structure. The compound names (and letter designation) are provided within the figure legends, rather than below each structural figure.

2.2) Many functional groups in chemical structures are distorted, especially acetoxy group. The authors should consult organic chemists for proper drawing.

The figures have been adapted and the functional groups are no longer distorted.

Reviewer 2 Report

The paper is in the field of the journal but the authors need to make some significant changes to improve the structure and content of the manuscript before it is published.

1. Please reduce the font size of the letters from the figures. They are to big in comparison to the compounds represented..

2. In my opinion Table 1 and Table 2 should be removed because the text from the tables overlap with the discussions.

3. The authors need to add more quantitative data regarding the information presented in section 4. In the current form it is a qualitative assessment that has very low importance if it is not accompanied by a quantitative assessment. Some compounds maybe found only in traces!?

4. Authors need to emphasize the novelty of the paper. I do not see what does someone gain in addition by reading this manuscript in comparison to other papers with similar subject.

5. Conclusion section needs to be revised and more numerical data added. Future perspectives need to be clearly defined.

6. List of abbreviations missing.

Author Response

The paper is in the field of the journal but the authors need to make some significant changes to improve the structure and content of the manuscript before it is published.

  1. Please reduce the font size of the letters from the figures. They are to big in comparison to the compounds represented.

The font size has now been revised for all figures.

  1. In my opinion Table 1 and Table 2 should be removed because the text from the tables overlap with the discussions.

We understand the reviewer’s concern as the tables take up substantial space in the manuscript. However, we believe that these tables are very important to highlight to the reader which Eremophila species should be focussed on for future research. We believe that these tables will greatly enhance citations of this study. Table 1 summarises the species that were used in traditional medicine, whilst Table 2 summarises the work published to date, for the purpose of highlighting gaps in the literature. We are reluctant to completely delete them as not all of the data presented in the tables is discussed in the text and we believe that they are important and relevant.

However, to address the reviewer’s concern:

  • We have moved the tables to supplementary data.
  • We have cited the tables within the text as “Supplementary Table 1” and “Supplementary Table 2”
  • We have also provided a reference section for the supplementary section.
  • The citation numbering has been altered in the tables, as multiple references were not also cited in the text of the manuscript and were therefore removed from the main manuscript.
  • This has resulted in multiple references no longer required in the body of the manuscript. Therefore, the in-text citation numbers and the reference section of the manuscript have also been substantially revised.

  1. The authors need to add more quantitative data regarding the information presented in section 4. In the current form it is a qualitative assessment that has very low importance if it is not accompanied by a quantitative assessment. Some compounds maybe found only in traces!?

We understand the reviewer’s point and we have addressed this in the revised manuscript (where possible) by providing the extraction yields of the individual compounds. This has resulted in numerous revisions in section 4. Where possible, we have provided the % yield of the compound extracted from the dried plant material. However, this was only available for approximately half of the studies reviewed in our manuscript. Multiple other studies have quantified the individual components as % relative abundance values. These are not ideal as they merely provide a measure of the area under the chromatographic peak(s) for the compounds of interest as % of the total area under all peaks. This method of quantification is not accurate and does not account for differences for each compound for each detection method (e.g. UV/Vis absorbance, MS etc). Therefore, % relative abundance is best regarded as a rough approximate value only. However, we acknowledge that quantification by % relative abundance is better than no estimate of the levels of the individual compounds. Therefore, we have included % relative abundance in the revised manuscript where it is available. For several studies, no quantitative estimate was provided. For those studies, we have stated that the authors did not specify the yield of the compound, to highlight that further studies are required to quantify the levels of these compounds in the extracts/essential oils.

  1. Authors need to emphasize the novelty of the paper. I do not see what does someone gain in addition by reading this manuscript in comparison to other papers with similar subject.

The reviewer is correct that there are several earlier reviews. These have been useful as a starting point and we have cited them within the manuscript. However, most of those studies are now outdated. Indeed, the important studies (Ghilsalberti, 1994; Richmond and Ghisalberti, 1994; Richmond, 1993) are each nearly 40 years old. Whilst several reviews have been published more recently, they have focused on specific aspects of Eremophila spp. e.g. the volatile components (Sandgrove et al, 2021); the antibacterial activity of terpenoid components (Biva et al, 2019) etc. There are no recent studies that review all of the therapeutic properties and the phytochemistry of Eremophila together in a single study. Furthermore, we are unaware of any studies that have reviewed ethnobotanical studies of Eremophila spp. use since 1994. Thus, we believe that this review is over-due and we believe that it will be useful to researchers in this field.

To address the reviewer’s comment and to make this clearer, we have added the following text on lines 124-131 of the marked-up manuscript:

“Several studies have previously reviewed the traditional; medicinal uses of Eremophila spp. and their phytochemistry, and these served as a good starting point for our study [4,5,6]. However, those reviews were published in 1993/1994 and are now outdated. Our study updates the earlier reviews to include the wealth of recent studies in this field. Or review focusses on the therapeutic properties of Eremophila spp. and on therapeutically important phytochemical constituents, with the aim of highlightling future studies into the medicinal potential of this important genus, and on its phytochemical constituents.”

  1. Conclusion section needs to be revised and more numerical data added. Future perspectives need to be clearly defined.

We disagree with the reviewer about the inclusion of numerical data in the conclusions section. We believe that numerical data should be included in the results section, and further elaborated on in the discussion section. The conclusions section of a manuscript should generally be reserved for reiterating the main argument of the study, and for summarising the findings and general trends, as well as indicating future directions. The conclusion section should not be a repetitive summary of the results.

With regards to the reviewer’s comment about future directions, we agree that the conclusion section should highlight future directions. We believe that we have already addressed this point in that section. For example:

(lines 734-737) However, despite the relative wealth of ethnopharmacological knowledge about the traditional usage of Eremophila species, rigorous scientific research has mainly focussed on a relatively few species, with the other species receiving substantially less attention.”

This highlights the need to study other related species.

(lines 744-752)However, for most species, the identification and mechanisms of action of the active principles have only been partially characterised and substantially more work is required. Despite these previous studies, most of the Eremophila species screened to date have only been tested for one or two therapeutic properties and screening against other bioactivities is required. At present, testing of Eremophila species for therapeutic properties has focussed on the species with a documented history of use in traditional medicinal systems. However, the taxonomic relationship between the Eremophila species suggests that testing other species of this genus for the same therapeutic properties that the more well-known species are used for is also warranted.”

This statement also highlights future work that is required.

  1. List of abbreviations missing.

We have followed the journal’s instructions to authors with regards to the abbreviations (see the text below, which was copied from the journal’s author instructions).

“Acronyms/Abbreviations/Initialisms should be defined the first time they appear in each of three sections: the abstract; the main text; the first figure or table. When defined for the first time, the acronym/abbreviation/initialism should be added in parentheses after the written-out form.”

Molecules requires that all abbreviations be defined when the first appear in the text, which we have already done. The abbreviations are not required to be listed separately and therefore, we had not done this. We have also searched through published Molecules papers and none have separate lists of abbreviations. Therefore, we believed that this point has already been addressed and we made no changes in response to this point. 

Round 2
